# Characterization of human exposure to *Anopheles* and *Aedes* bites using antibody-based biomarkers in rural zone of Cameroon

Idriss Nasser Ngangue-Siewe[1,2,3]*, Paulette Ndjeunia-Mbiakop[2,4], André Barembaye Sagna[5], Abdoul-Aziz Mamadou Maïga[3], Roland Bamou[2,6], Antoine Sanon[3], Jeannette Tombi[4], Jean Arthur Mbida Mbida[1], Christophe Antonio-Nkondjio[2,7], Franck Remoue[5]*, Athanase Badolo[3]

1 Faculty of Science, Department of Animal Biology and Physiology, The University of Douala, Douala, Cameroon, 2 Malaria Research Laboratory, Organisation de Coordination pour la Lutte Contre les Endémies en Afrique Centrale (OCEAC), Yaoundé, Cameroon, 3 Laboratory of Fundamental and Applied Entomology, University Joseph Ki-Zerbo, Ouagadougou, Burkina Faso, 4 Faculty of Science, Department of Animal Biology and Physiology, The University of Yaoundé, Yaoundé, Cameroon, 5 Institut de Recherche Pour le Développement (IRD), MIVEGEC Unit, University of Montpellier, IRD, CNRS, DR Occitanie, Montpellier, France, 6 Laboratory of Malaria and Vector Research, National Institute of Allergy and Infectious Diseases (NIAID), National Institute of Health (NIH), Rockville, Maryland, United States of America, 7 Vector Biology, Liverpool School of Tropical Medicine, Pembroke Place, Liverpool, United Kingdom

* idrissngangue@gmail.com (INN-S); franck.remoue@ird.fr (FR)

**Data Availability Statement:** All relevant data are within the paper and its Supporting Information files.

## Abstract

Malaria and *Aedes*-borne diseases remain major causes of mortality, morbidity, and disability in most developing countries. Surveillance of transmission patterns associated with vector control remains strategic for combating these diseases. Due to the limitions of current surveillance tools used to assess human exposure to mosquito bites, human antibody (Ab) responses to salivary peptides from *Anopheles* (gSG6-P1) and *Aedes* (Nterm-34kDa) are increasingly being used to measure direct human-*Anopheles* or *Aedes* contact. This study reports on the assessment of Human IgG Ab responses to gSG6-P1 and Nterm–34-kDa salivary peptides as biomarkers to track exposure to *Anopheles* and *Aedes* bites, in rural localities of Cameroon. Blood samples were collected between October and November 2022 from 173 individuals residing in four villages: Njombe, Kekem, Belabo, and Ouami. Sociodemographic characteristics and information regarding Long Lasting Insecticide Net (LLIN) ownership, use, and net characteristics were recorded using a questionnaire. The measurement of human IgG levels to gSG6-P1 and Nterm-34kDa peptides was conducted in blood samples using ELISA. The levels of IgG responses to *Anopheles* gSG6-P1 and *Aedes* Nterm-34kDa salivary peptides varied significantly across villages (all p<0.05). IgG responses to *Anopheles* gSG6-P1 were higher in Njombe compared to Belabo and Ouami (all p<0.01), while IgG responses to *Aedes* Nterm-34kDa were higher in Kekem compared to the other villages (all p<0.0001). Aweak correlation was observed between IgG responses to *Anopheles* and *Aedes* salivary peptides (Spearman r = 0.2689, p = 0.0003). However, the median level of IgG to *Anopheles* gSG6-P1 was higher than IgG to *Aedes* Nterm-34kDa in Njombé, Belabo, and Ouami. Individuals not using their LLIN, those using damaged bed nets, and those who reported vegetation around their houses developed

**Funding:** This work received financial support from the Bill & Melinda Gates Foundation, Grant ID: OPP1210340 to C.A.-N and Pan African Mosquito Control (PAMCA) association for the field work and from "Appui à la Formation, la Recherche et à l'Innovation pour le Développement Intra-Afrique" (AFRIDI) fellowship under Intra-Africa Academic Mobility Scheme of the European Union for the laboratory analysis to I.N.N.-S. the funders had no role in study design, data collection and analysis, decision to publish, or preparation of the manuscript.

**Competing interests:** The authors have declared that no competing interests exist.

higher IgG responses to gSG6-P1 and Nterm–34 kDa compared to those who did not (all p<0.05). The immune-epidemiological biomarkers have shown promising potential as indicators for monitoring human exposure to various mosquito bites and their heterogeneity in the same site. However, additional research is needed to validate the efficacy of this technique for surveillance purposes and to assess the effectiveness of vector control interventions.

## Introduction

*Anopheles* and *Aedes* mosquitoes are vectors of significant infectious diseases including malaria, dengue, chikungunya, and yellow fever. Globally, malaria affect over 240 million poeple and cause more than 600,000 deaths annually [1]. In Cameroon, the prevalence of this disease ranges from 24 to 30%, with children under 5 years and pregnant women being the most affected groups [1]. Among arboviruses, dengue fever (DF) is a critical arboviral disease affecting approximately 3.9 billion people in 128 countries, representing 40–50% of the global population [2]. Approximatively, 400 million cases of DF occur each year, leading to 20 to 25,000 deaths, affecting predominantly children in developing nations [3]. Over the past decade, several epidemics of dengue, chikungunya, and yellow fever outbreaks have been reported in sub-Saharan Africa [4–6]. In Cameroon, numerous cases of dengue, chikungunya, and yellow fever have been frequently reported since 2006 [5,7–9].

In order to advance towards the elimination of malaria and arboviral diseases, surveillance activities are crucial to inform policy and for evidence-based decision-making. Sampling techniques commonly used to measure exposure to *Anopheles* and *Aedes* bites include mosquito collection using various methods or human landing catches. While these methods provide valuable information to monitor vector-human interactions, they have limitations such as dexterity at the individual level, cost, logistical challenges, and ethical concerns [10–12]. These limitations imply that they may not be effective in all epidemiological contexts and could introduce certain biases. Additional tools, such as biomarkers of human exposure to *Anopheles* [13] and *Aedes bites* [14], have been developed and validated to quantitatively and individually measure the level of exposure of human populations to malaria and arboviral vectors. These biomarkers are based on assessing the level of IgG antibody (Ab) response to proteins/peptides from mosquito saliva. Studies conducted so far indicate that human IgG levels to *Aedes* Nterm -34Da protein and *Anopheles* gSG6-P1 are specific biomarkers highly conserved between species and are particularly relevant for monitoring exposure to arboviral and malaria vectors, even in a context of low exposure to these mosquito bites [15–18].

The aim of the present study was to explore the heterogeneity of human exposure to both *Aedes* and *Anopheles* bites in four rural settings in Cameroon by using both biomarkers and the potential impact of sociodemographic factors.

## Material and methods

### Study sites

A cross-sectional study was conducted from October to November 2022 in four localities: Ouami, Belabo, Kékem, and Njombé (Fig 1). Ouami (5˚16'60"N, 13˚34'60" E) and Belabo (4˚56'00"N, 13˚18'00" E) are situated in the East Forest region of Cameroon near the Lom-Pangar hydroelectric dam along the Sanaga river. The area is known for its floodplains extensively

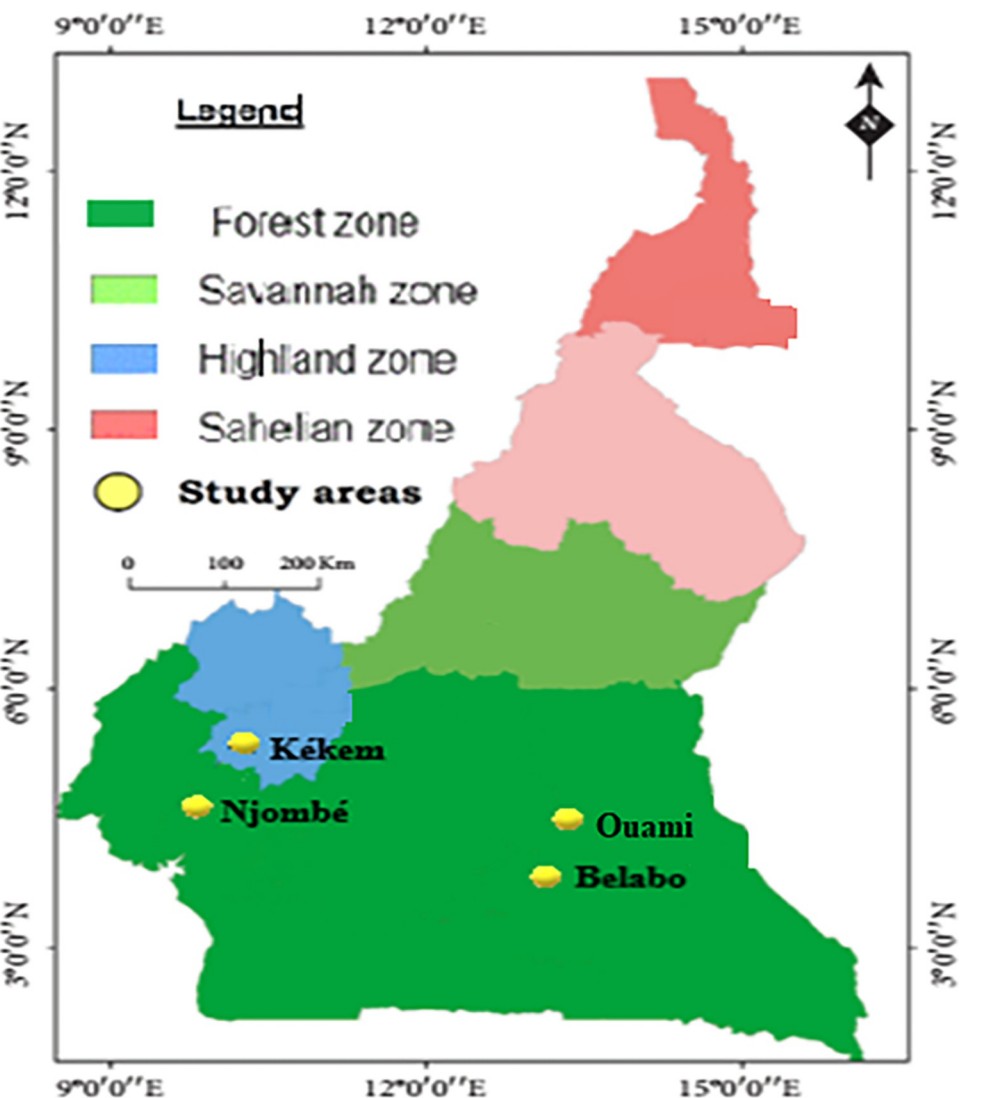

**Fig 1. Map of Cameroon showing the study sites (Published by Ngangue-Siewe *et al.*, 2022) [25].**

used for fishing. The climate is equatorial with four seasons: a long dry season from mid-November to mid-March, a short rainy season from mid-March to June, a short dry season from July to August, and a long rainy season from September to mid-November. Kekem (5° 10'00"N, 10°02'00"E) is located in the West Region at the base of the mountains. The climate in this area is characterized by two seasons: a dry season from November to March and a rainy season from April to October. Njombé (4°64'65″ N, 9°67'83″ E) is situated in the Littoral Region, characterized by a nine-month long rainy season (March to November) and a short dry season (December to February).

## Ethical clearance

The study received approval from the National Human Health Research Ethics Committee of Cameroon under No. 2020/04/1209/CE/CNERSH/SP, and administrative authorization was obtained from the local divisional officer. Oral informed consent was obtained from adult participants, as well as parents or guardians of children under 18 years of age. This verbal

agreement was noted on each individual survey form and was approved by the Institutional Review Board. Participants included in the study had resided in each locality for at least one month. In appreciation of their voluntary participation, individuals showing symptoms of common illnesses were provided with appropriate medications.

## Data collection

The selection of households was done randomly, and interviews were carried out in each household using a structured questionnaire to collect socio-demographic information, age, gender, ownership, and utilization of LLINs. The presence of vegetation around the house was recorded through visual inspection. If they agreed to participate, up to three individuals were sampled from each household. Blood samples were collected on Whatman 3 MM paper using the dried blood spot (DBS) technique and stored at 4˚C until needed.

**Salivary peptides gSG6-P1 and Nterm–34-kDa.**   Synthetic forms of the antigenic peptides gSG6-P1 (Catalogue number GPS_1216, Genepep, Saint Jean de Vedas, France) and Nterm-34–kDa (Catalogue number GPS_2958, Genepep, Saint Jean de Vedas, France) were each resuspended in ultra-filtered water and stored at a concentration of 1 mg/mL at -20˚C until use.

**Assessment of human IgG antibody levels against gSG6-P1 and Nterm–34-kDa.**   DBSs (diameter, 0.8 cm) were eluted as previously described [14]. ELISA assays were carried out on DBS eluates separately to assess IgG responses to gSG6-P1 and Nterm–34-kDa salivary antigens following established protocols [14,16]. Briefly, Maxisorp plates (Nunc, Roskilde, Denmark) were coated with 20 μg/ml of Nterm–34-kDa or gSG6-P1 and incubated at 37˚C for 2 hours and 30 minutes. The plates were then blocked for 1 hour at room temperature using 300 μL of protein-free blocking buffer (Thermoscientific, Rockford, United States). Eluates diluted at 1/20 in PBS-Tween 1% were added and incubated overnight at +4˚C. Biotinylated mouse anti-human IgG (BD Pharmingen, San Diego CA, USA) was subsequently added at a concentration of 1/1000 in PBS-Tween 1% to detect bound human IgG, followed by the addition of streptavidin-conjugated peroxidase (GE Healthcare, Orsay, France) at 1/1000 in PBS-Tween 1%. ABTS (2,2'-azino-bis (3 ethylbenzthiazoline 6-sulfonic) diammonium acid; Sigma, St Louis, MO, USA) was used as the substrate, and optical densities (ODs) were measured at 405 nm after 2 hours of development. Each sample was assayed in duplicate wells containing salivary peptide and in a well without antigen to account for non-specific reactions. The individual results were quantified as the $\Delta$OD value: $\Delta$OD = ODx − ODn, where ODx represents the mean of the individual OD values in both wells with salivary antigen and ODn represents the individual OD value in a blank well without antigen. The reproducibility between ELISA plates has been verified by using 3 positive controls (low, medium, high of specific IgG levels) in each plate to monitor plate to plate variations. However, the study faced difficulty in using a negative control to calculate the cut-off due to challenge in obtaining serum from unexposed individuals in Africa.

**Statistical analysis.**   All data were analyzed using Graph Pad Prism® software (Graph Pad Software, San Diego, California, United States) version 9. Spearman correlation analysis was employed to compare IgG antibody levels against *Anopheles* and *Aedes* salivary antigens. After verifying that the data did not follow a Gaussian distribution or Normality test, the comparison between two different groups (quantitative variables) was conducted using the non-parametric Mann-Whitney test. Comparisons between multiple groups were performed with the non-parametric Kruskal-Wallis tests (for independent series). Dunn's post-test was utilized for multiple paired comparisons between villages. Significance was determined at a p-value < 0.05.

## Results

### Characteristics of the study population by village

A total of 173 study participants were enrolled in the four villages: 55 individuals in Njombé, 40 in Kekem, 51 in Belabo, and 27 in Ouami (Table 1). Participants were evenly distributed among four age groups (0 to 5 years, 6 to 10 years, 11 to 15 years, and over 15 years). However, there was a slight underrepresentation of children aged 11–15 years in Njombé and those aged 0–5 years in Belabo. The sex ratio favored females (2:1) in Njombe, while it favored males in Ouami (Table 1). The majority of participants (>90%) owned a LLIN, but not all (41.8–85%) reported using them. Most of the nets were damaged with holes in Njombé (69.8%), Belabo (71.7%), and Ouami (90%). Njombé village had the lowest number of participants using LLINs (41.8%) despite a high ownership rate. Kekem recorded the highest number of participants owning (97.5%) and using (85%) LLINs, and the difference was significant after a group comparison with other sites (P = 0.006; P = 0.0001 for those two variables respectively). In all the study villages, LLINs were used mostly every night. Regarding the presence of vegetation around the houses, in all localities, the majority of houses had vegetation around them except in Belabo (Table 1).

### IgG levels against *Anopheles* and *Aedes* salivary peptides

Globally, the specific IgG levels ranged from 0.011 and 1.073 for the Nterm-34-kDa peptide and from 0.005 to 1.185 for the gSG6-P1 peptide. The comparison for both peptides was

**Table 1. Socio-demographic characteristics of study population by village of residence.**

| Variable | Njombé (N = 55) | Kekem (N = 40) | Belabo (N = 51) | Ouami (N = 27) | p-value (Chi-squared) |
|---|---|---|---|---|---|
| **Sex** | | | | | |
| Female | 37 (67.3) | 19 (47.5) | 21 (41.2) | 09 (33.3) | 0.010 (11.2465) |
| Male | 18 (32.7) | 21 (52.5) | 30 (58.8) | 18 (66.67) | |
| **Age (in year)** | | | | | |
| 0–5 | 18 (32.7) | 10 (25) | 07 (13.7) | 06 (22.2) | 0.439 (8.9755) |
| 6–10 | 17 (30.9) | 09 (22.5) | 19 (37.3) | 06 (22.2) | |
| 11–15 | 08 (14.6) | 10 (25) | 12 (23.5) | 08 (29.6) | |
| >15 | 12 (21.8) | 11 (27.5) | 13 (25.5) | 07 (26) | |
| **LLIN ownership** | | | | | |
| No | 02 (03.6) | 01 (02.5) | 05 (09.8) | 07 (25.93) | 0.003 (13.9184) |
| Yes | 53 (96.4) | 39 (97.5) | 46 (90.2) | 20 (74.07) | |
| **LLIN use** | | | | | |
| No | 32 (58.2) | 06 (15) | 09 (17.6) | 13 (48.2) | <0.0001 (28.9421) |
| Yes | 23 (41.8) | 34 (85) | 42 (82.4) | 14 (51.8) | |
| **LLIN with holes** | | | | | |
| No | 16 (30.2) | 36 (92.3) | 13 (28.3) | 02 (10) | <0.0001 (55.3715) |
| Yes | 37 (69.8) | 03 (07.7) | 33 (71.7) | 18 (90) | |
| **Period of LLIN usage** | | | | | |
| Every night | 46 (86.8) | 39 (100) | 46 (100) | 20 (100) | 0.002 (14.5108) |
| Rainy season | 07 (13.2) | 0 (0) | 0 (0) | 0 (0) | |
| **Vegetation around the house** | | | | | |
| No | 25 (45.5) | 5 (12.5) | 27 (52.9) | 11 (40.7) | 0.0007 (16.9175) |
| Yes | 30 (54.5) | 35 (87.5) | 24 (47.1) | 16 (59.3) | |

N = total number of participants.

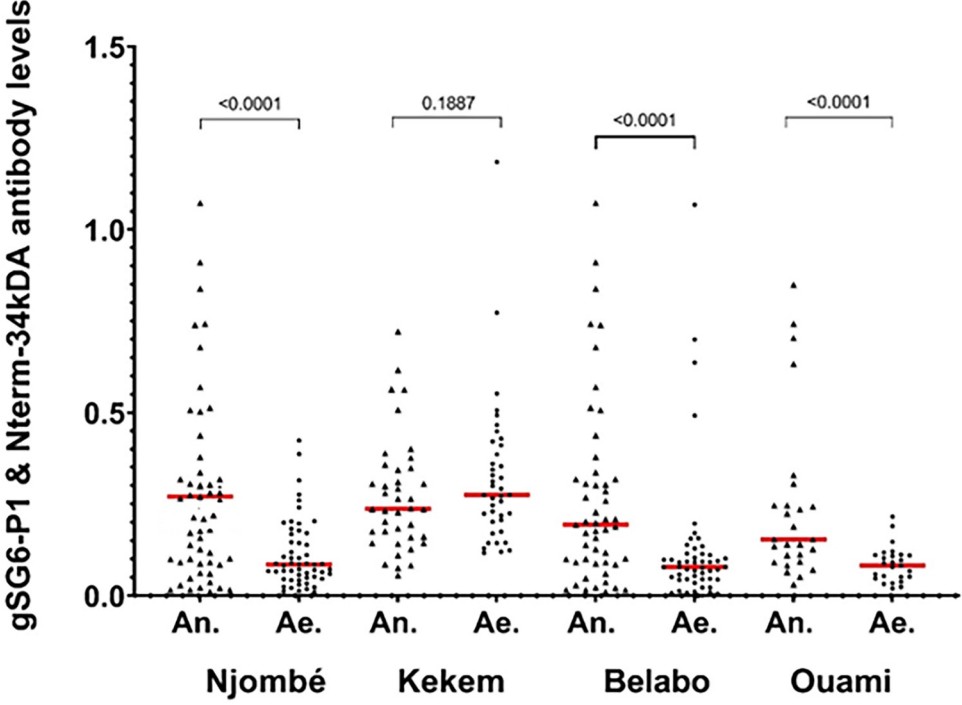

**Fig 2. Comparison between gSG6-P1 IgG levels and Nterm-34kDa IgG levels by village.** Triangular and round dots indicate individual IgG responses to gSG6-P1 and Nterm-34kDa salivary peptides, respectively, while red bars represent median values in each village. Statistically significant differences between all paired antibody levels (Wilcoxon paired test) are indicated. An. = *Anopheles*; Ae. = *Aedes*.

conducted by village of residence, and, except for Kekem village where there was no significant difference between IgG responses to both salivary peptides, IgG responses towards *Anopheles* mosquito bites were higher than those towards *Aedes* mosquitoes in Njombé, Belabo, and Ouami (p<0.0001) (Fig 2). We used Kruskal-Wallis test when analyzing exposure to *Anopheles* and *Aedes* bites across villages and both IgG levels to the gSG6-P1 and Nterm-34kDa varied significantly. IgG levels to the gSG6-P1 were significantly higher in Njombé compared to Kekem, Belabo, and Ouami (p = 0.01), while IgG levels to the Nterm-34kDa were higher in Kekem compared to the other three other villages (p<0.0001) (Fig 2 and Table 2).

## Sociodemographic factors influencing exposure to *Anopheles* and *Aedes*

IgG levels specific to *Anopheles* gSG6-P1 and *Aedes* Nterm–34 kDa salivary peptides were analyzed according to the village of residence, gender, age groups, bed net ownership, bed net use, bed net condition, and presence of vegetation around the house (Table 2). Age groups, gender, and bed net ownership did not appear to influence the IgG levels to *Anopheles* and *Aedes* salivary peptides in the study area (all p>0.05). However, bed net use, bed net condition, and the presence of vegetation around the house significantly impacted the IgG responses to *Anopheles* and *Aedes* salivary peptides. Participants who reported using their bed nets had notably lower median IgG responses to the *Anopheles* gSG6-P1 (p<0.0076) than those who reported not using them. Surprisingly, the opposite effect was observed for *Aedes* exposure, as participants who reported using their bed nets had significantly higher median IgG responses to the *Aedes* Nterm -34 kDa (p<0.0005) than those who reported not using their bed nets. This situation was specific to Njombé (p = 0.0069) (S1 File).xs

**Table 2. Socio-demographic factors influencing the exposure to *Anopheles* and *Aedes* mosquitoes.** Mann-Whitney test was used when there are two groups and Kruskal-Wallis test when there are more than two groups.

| Variable | Exposure to *Anopheles* | | | | Exposure to *Aedes* | | | |
|---|---|---|---|---|---|---|---|---|
| | No. | Median | 25th–75th Percentile | *p*-value | No. | Median | 25th–75th Percentile | *p*-value |
| **Village** | | | | | | | | |
| Njombé | 55 | 0.286 | 0.183–0.459 | 0.0090 | 55 | 0.085 | 0.049–0.168 | <0.0001 |
| Kekem | 40 | 0.237 | 0.161–0.348 | | 40 | 0.276 | 0.203–0.405 | |
| Belabo | 51 | 0.194 | 0.091–0.318 | | 51 | 0.079 | 0.045–0.114 | |
| Ouami | 27 | 0.154 | 0.093–0.247 | | 27 | 0.083 | 0.047–0.111 | |
| **Age** | | | | | | | | |
| 0–5 | 41 | 0.216 | 0.128–0.382 | 0.5204 | 41 | 0.119 | 0.060–0.236 | 0.2418 |
| 6–10 | 50 | 0.237 | 0.101–0.382 | | 50 | 0.084 | 0.048–0.144 | |
| 11–15 | 42 | 0.270 | 0.160–0.504 | | 42 | 0.111 | 0.054–0.260 | |
| >15 | 40 | 0.219 | 0.144–0.319 | | 40 | 0.103 | 0.0780.206 | |
| **Sex** | | | | | | | | |
| Female | 86 | 0.235 | 0.120–0.352 | 0.6655 | 86 | 0.105 | 0.067–0.202 | 0.5778 |
| Male | 87 | 0.244 | 0.142–0.379 | | 87 | 0.098 | 0.052–0.204 | |
| **LLIN ownership** | | | | | | | | |
| Yes | 158 | 0.241 | 0.137–0.380 | 0.2843 | 158 | 0.103 | 0.058–0.203 | 0.3249 |
| No | 15 | 0.172 | 0.128–0.302 | | 15 | 0.098 | 0.054–0.111 | |
| **LLIN use** | | | | | | | | |
| Yes | 113 | 0.209 | 0.119–0.318 | 0.0076 | 113 | 0.121 | 0.070–0.253 | 0.0005 |
| No | 60 | 0.290 | 0.159–0.498 | | 60 | 0.072 | 0.046–0.112 | |
| **LLIN with holes** | | | | | | | | |
| Yes | 67 | 0.269 | 0.175–0.468 | 0.0170 | 67 | 0.204 | 0.091–0.354 | <0.0001 |
| No | 91 | 0.216 | 0.102–0.327 | | 91 | 0.079 | 0.046–0.118 | |
| **Vegetation around the house** | | | | | | | | |
| Yes | 105 | 0.269 | 0.169–0.451 | <0.0001 | 105 | 0.157 | 0.092–0.284 | <0.0001 |
| No | 68 | 0.172 | 0.095–0.291 | | 68 | 0.061 | 0.024–0.089 | |

As anticipated, participants who reported using bed nets with holes or those with noted vegetation around their homes exhibited significantly higher median levels of IgG towards both *Anopheles* (median = 0.269 and 0.269, respectively) and *Aedes* (median = 0.204 and 0.157, respectively) compared to those who did not (p<0.05) (Table 2).

For the use of mosquito nets in poor condition (with holes) versus good condition (no holes) per village, the difference was significant only for exposure to *Anopheles* bites in Bélabo (median good condition = 0.158, median poor condition = 0.438, p = 0.005). However, this difference was not significant for exposure to *Aedes* (Kekem: p = 0.860; Bélabo: p = 0.099; Njombe: p = 0.966; Ouami: p = 0.947; Ouami: p = 0.937) and *Anopheles* (Kekem: p = 0.274; Njombe: p = 0.719; Ouami: p = 0.937) in the other three other sites.

Concerning the presence of vegetation around households per village, there was no significant difference in the median level of IgG responses to the *Anopheles* peptide (Kekem: p = 0.793; Belabo: p = 0.221; Njombe: p = 0.442; Ouami: p = 0.999) and *Aedes* peptide (Kekem: p = 0.170; Belabo: p = 0.145; Njombe: p = 0.609; Ouami: p = 0.860) across all sites.

## Correlation between IgG levels against *Anopheles* (gSG6-P1) and *Aedes* (Nterm-34kDa) salivary antigens

We further investigated the correlation between IgG levels against *Anopheles* (gSG6-P1) and *Aedes* (Nterm-34kDa) salivary antigens using blood samples from the same individuals. A

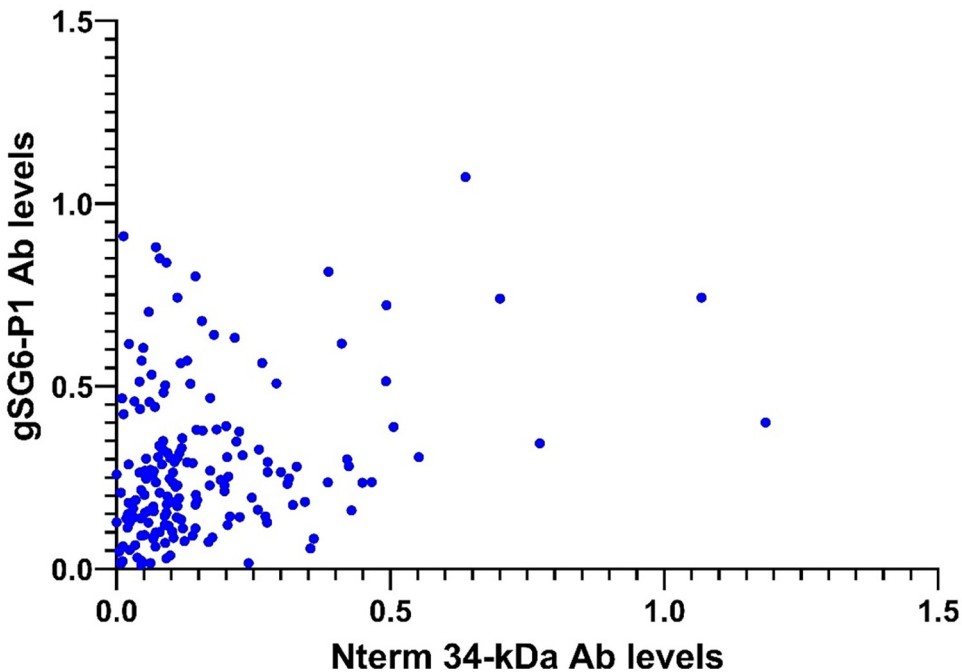

**Fig 3. Comparison of exposure to *Anopheles* (antibody responses to gSG6-P1) and *Aedes* (antibody responses to Nterm-34) mosquitoes in the same individual.**

weak but statistically significant correlation was observed (Spearman r = 0.2689 (95% CI: 0.1203–0.4057), p = 0.0003) (Fig 3). Upon village-specific analysis, the correlation of IgG levels against *Anopheles* (gSG6-P1) and *Aedes* (Nterm-34kDa) salivary antigens was significant within the same individuals in the villages of Bélabo (Spearman r = 0.450, CI (0.190–0.650); p = 0.001) and Ouami (Spearman r = 0.494, CI (0.129–0.741), while it was not significant in Kekem (Spearman r = 0.220, CI (-0.108–0.504); p = 0.173) and Njombé (Spearman r = 0.028, CI (-0.247–0.298); p = 0.840).

## Discussion

The study assessed IgG responses against *Anopheles* gSG6-P1 and *Aedes* Nterm-34 kDa salivary peptides, which serve as indicators of human exposure to *Anopheles* and *Aedes* mosquito bites, in individuals residing in four distinct rural areas in Cameroon. The findings revealed varying patterns of human exposure to bites from both *Anopheles* and *Aedes* mosquitoes. In Njombé, Bélabo, and Ouami, the median IgG responses to *Anopheles* gSG6-P1 were notably higher than those to *Aedes* Nterm-34kDa, whereas in Kekem, the median IgG levels to Nterm-34kDa were higher compared to the other three villages. This diversity in human exposure to *Anopheles* and *Aedes* bites across villages may stem from different ecological and environmental factors. Factors such as the presence of various habitat types like man-made habitats, agricultural activities, proximity to water bodies, water storage containers, varied climatic conditions, and different human behaviours were identified as contributing to the creation of suitable habitats for different mosquito species. These factors could increase the risk of human exposure to both *Anopheles* and *Aedes* mosquito bites [19,20]. Previous studies in Cameroon using the gSG-P1 biomarker have shown differences in biomarker expression levels between mainland and island populations, indicating varying transmission risks across the country [21]. Similar observations were reported in previous studies in Senegal [20]. *Aedes* mosquitoes,

particularly *Ae. albopictus* and *Ae. aegypti*, are prevalent in many regions in the south of Cameroon [22,23]. The lower exposure to *Aedes* mosquito bites in Njombé, Bélabo, and Ouami may be attributed to seasonal fluctuations. Conversely, the higher exposure to both *Aedes* and *Anopheles* mosquitoes in Kekem could result from a favourable environment for the proliferation of both vectors such as the exploitation of lowland areas for the practice of seasonal gardening activities, widespread presence of water storage containers in households for rainwater collection, and the abundance of old tires serving various purposes or left in the environment.

Regarding the distribution of *Anopheles* species, previous studies have shown the presence of both *An. gambiae* ss and *An. coluzzii* in Belabo and Ouami, while *An. coluzzi* and *An. gambiae* ss were identified as the predominant species in Njombe and Kekem respectively [24,25].

The potential influence of sociodemographic characteristics, human behavior, and living environment was compared to the level of exposure to *Anopheles* and *Aedes* bites. Analyses of IgG responses against *Anopheles* and *Aedes* salivary peptides showed no significant differences according to age, gender, and possession of LLINs. The study findings do not align with previous reports which indicated an increase in exposure to mosquito bites with age in Senegal [13,20,26].

Individual IgG responses to *Anopheles* gSG6-P1 and *Aedes* Nterm-34kDa salivary peptides were also analysed based on LLIN ownership, usage rate, and net physical status. This analysis revealed no significant difference in IgG responses to both salivary peptides with bed net ownership which could be associated with the high ownership rate of bed nets in the country. However, the frequency of bed net usage and the physical condition of the nets (presence of holes) were significantly associated with IgG responses to *Anopheles* gSG6-P1 and *Aedes* Nterm-34kDa salivary peptides. Individuals who reported sleeping under bed nets every night had notably lower IgG responses to *Anopheles* gSG6-P1, while bed net usage did not correlate with IgG responses to *Aedes* Nterm-34-kDa. This finding aligns with previous research [18,21,26,27]. The levels of specific IgG in the participants varied significantly based on the physical condition of the LLINs, with higher levels observed in individuals using damaged LLINs compared to those using intact ones. Similar trends were reported in prior studies, where individuals using coils or spray bombs had lower IgG responses to gSG6-P1 compared to non-users [21]. This pattern is consistent with research from Benin [28] and Ivory coast [18], demonstrating that the use of LLINs in good condition is associated with reduced exposure to *Anopheles* mosquito bites. These findings highlight the relevance and sensitivity of this biomarker not only for assessing LLIN efficacy but also for evaluating the physical integrity of LLINs, as previously shown by Noukpo in Benin [28].

The presence of vegetation around houses was found to increase exposure to *Aedes* bites. People living close to vegetation had higher levels of IgG responses to gSG6-P1 and Nterm–34 kDa compared to those living in houses with little vegetation around their home. This result was consistent with findings from various studies conducted elsewhere [20,29].

Since the study was conducted in October during the rainy season, it is possible that seasonal fluctuations were not fully captured, warranting further attention. Additionally, several other individual factors such as migration of individuals across villages, use of other vector control strategies (e.g., coils, indoor residual spraying), genetic predispositions, and levels of education/awareness of malaria control strategies, which could potentially influence the immunological results of our study have not been assessed. Future studies may be needed to evaluate their potential impacts.

The study underscores the importance of integrating various sampling methods to enhance the surveillance of both malaria and arbovirus vectors. Additionally, it highlights the necessity of implementing an integrated vector management program that considers the spatial heterogeneity of transmission risk to maximize the effectiveness of interventions in the field.

## Conclusion

Human antibody responses towards *Anopheles* gSG6-P1 and *Aedes* Nterm-34kDa salivary peptides varied depending on the village of residence, bed net usage and condition, and the presence of vegetation around houses. While more individuals were exposed to *Anopheles* bites compared to *Aedes* bites, those highly exposed to *Anopheles* are not necessarily highly exposed to *Aedes* mosquitoes, and vice versa, suggesting the heterogeneity of exposure, at the individual level, to major mosquito species in specific settings. These immunological tools seem to be therefore valuable for informing mosquito-borne disease control programs for evaluating vector control strategies on human-vector contact and adjusting control strategies based on exposure variation. However, standardized and optimized methods for assessing these specific antibody-based biomarkers (e.g., through the development of Rapid Diagnostic Tests) are essential the operational utilization of these tools by vector control and surveillance programs.

## Supporting information

**S1 File. Comparison of gSG6-P1 antibody levels and Nterm-34kDa antibody levels according to bed net use by village.** Red bars represent median values in each village.
(PDF)

**S2 File. Data base with Delda DO for *Anopheles* and *Aedes*.**
(XLSX)

## Acknowledgments

We are grateful to the administrative and traditional authorities, to Chi Nji Princewill and the population of Njombé, Kékem, Belabo and Ouami for their participation in the study.

## Author Contributions

**Conceptualization:** Idriss Nasser Ngangue-Siewe, Christophe Antonio-Nkondjio, Franck Remoue, Athanase Badolo.

**Data curation:** Idriss Nasser Ngangue-Siewe, André Barembaye Sagna, Franck Remoue.

**Formal analysis:** Idriss Nasser Ngangue-Siewe, André Barembaye Sagna, Abdoul-Aziz Mamadou Maïga, Franck Remoue.

**Funding acquisition:** Idriss Nasser Ngangue-Siewe, Christophe Antonio-Nkondjio.

**Investigation:** Idriss Nasser Ngangue-Siewe, Paulette Ndjeunia-Mbiakop.

**Methodology:** Idriss Nasser Ngangue-Siewe, Paulette Ndjeunia-Mbiakop, André Barembaye Sagna, Abdoul-Aziz Mamadou Maïga, Franck Remoue.

**Supervision:** Jean Arthur Mbida Mbida, Christophe Antonio-Nkondjio, Athanase Badolo.

**Validation:** Idriss Nasser Ngangue-Siewe, Paulette Ndjeunia-Mbiakop, André Barembaye Sagna, Roland Bamou, Jeannette Tombi, Jean Arthur Mbida Mbida, Christophe Antonio-Nkondjio, Franck Remoue, Athanase Badolo.

**Visualization:** Idriss Nasser Ngangue-Siewe, André Barembaye Sagna, Roland Bamou, Franck Remoue, Athanase Badolo.

**Writing – original draft:** Idriss Nasser Ngangue-Siewe.

**Writing – review & editing:** Paulette Ndjeunia-Mbiakop, André Barembaye Sagna, Abdoul-Aziz Mamadou Maïga, Roland Bamou, Antoine Sanon, Jeannette Tombi, Jean Arthur Mbida Mbida, Christophe Antonio-Nkondjio, Franck Remoue, Athanase Badolo.

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
