## [Decision Letter · Decision Letter 0]

4 Sep 2024

PONE-D-24-31144Characterization of human exposure to Anopheles and Aedes bites using antibody-based biomarkers in rural zone of CameroonPLOS ONE

Dear Dr. Ngangue-Siewe,

Thank you for submitting your manuscript to PLOS ONE. After careful consideration, we feel that it has merit but does not fully meet PLOS ONE’s publication criteria as it currently stands. Therefore, we invite you to submit a revised version of the manuscript that addresses the points raised during the review process.

We look forward to receiving your revised manuscript.

Kind regards,

Nazarudin Safian, PhD

Academic Editor

PLOS ONE

Journal Requirements:

2. In the ethics statement in the Methods, you have specified that verbal consent was obtained. Please provide additional details regarding how this consent was documented and witnessed, and state whether this was approved by the IRB.

4. Thank you for stating the following financial disclosure: This work received financial support from the Bill & Melinda Gates Foundation, Grant ID: OPP1210340 to C.A.-N and Pan African Mosquito Control (PAMCA) association for the field work and from “Appui à la Formation, la Recherche et à l’Innovation pour le Développement Intra-Afrique” (AFRIDI) fellowship under Intra-Africa Academic Mobility Scheme of the European Union for the laboratory analysis to I.N.N.-S.   

6. We note that Figure 1 in your submission contain [map/satellite] images which may be copyrighted. All PLOS content is published under the Creative Commons Attribution License (CC BY 4.0), which means that the manuscript, images, and Supporting Information files will be freely available online, and any third party is permitted to access, download, copy, distribute, and use these materials in any way, even commercially, with proper attribution. For these reasons, we cannot publish previously copyrighted maps or satellite images created using proprietary data, such as Google software (Google Maps, Street View, and Earth). For more information, see our copyright guidelines: http://journals.plos.org/plosone/s/licenses-and-copyright.

Reviewers' comments:

Reviewer's Responses to Questions

**Comments to the Author**

1. Is the manuscript technically sound, and do the data support the conclusions?

Reviewer #1: Partly

Reviewer #2: Partly

2. Has the statistical analysis been performed appropriately and rigorously? 

Reviewer #1: No

Reviewer #2: No

3. Have the authors made all data underlying the findings in their manuscript fully available?

Reviewer #1: No

Reviewer #2: Yes

4. Is the manuscript presented in an intelligible fashion and written in standard English?

Reviewer #1: No

Reviewer #2: Yes

5. Review Comments to the Author

Reviewer #1: PONE-D-24-31144

Review comments for the author.

The objective of this study was to utilize a serological approach to determine the level of exposure and heterogeneity of Anopheles and Aedes bites in four villages in Cameroon. This study is relevant since mosquito borne diseases are a public health concern in Cameroon.

Experimental approach

• The study utilized ELISA to draw its conclusion. An alternative tool should have been used alongside this method in order to compare and validate these findings. In the current state, the data provided does not provide reasonable information to conclude whether this method is better than conventional methods that have been previously used (refer to introduction section lines 76–81).

• The authors did not use appropriate controls in their assay. Although the reported values are changes in OD levels, samples from positive and negative individuals (exposed and non-exposed individuals) should have been included in the assay. Similarly, the cut-off value was not determined or indicated.

• The study observed variation in OD values with regards to LLIN use. These antigens can last up to 40 days before they begin to wane. A follow-up study would be significant in determining this variation and would strengthen the conclusion.

• Additional confounding factors were not factored in. For example, the authors do not show how they controlled for the migration of individuals across villages during the sampling period. The use of other vector control strategies like coils, indoor residual spraying, level of education/awareness of malaria control strategies, etc., at the household level was not considered, and this might influence the outcome of the study.

Data analysis

• The authors should justify why they opted for a non-parametric test instead of a parametric test.

• The authors should consider using a violin plot or box plot in Figure 2.

• What statistical test is represented in Table 2? Is it the post-hoc test or Kruskal-Wallis test? This should be reflected in line 182.

• Supplementary File 1 should also include a comparison of the peptides with other variables such as age, gender, and the condition of bed nets.

• The correlation between the two peptides should be compared in each village, as opposed to clumping everything together (Figure 3).

• There is no evidence of the data reported on the correlation of village-specific analysis (lines 224–29).

• Supplementary data with all OD levels would be useful.

Study conclusion.

The study, though relevant, is not novel. The correlation between these findings and the infection rate would be more relevant. qPCR can be used to determine parasitemia levels in these samples.

Reviewer #2: In general, I am supportive of the manuscript "Characterization of human exposure to Anopheles and Aedes bites using antibody-based biomarkers in rural zone of Cameroon". I believe that such field studies that try to understand that general exposure of populations to vectors of disease are vital pieces of information in the effort of implementation of control measures. I appreciate the author's effort to collect demographic information about their study groups and to inform the reader about topographic differences between locations, which are vital pieces of information to the study. However, I do not think that the authors made the most of this data.

1.) The demographic data presented in table 1 offers itself for comparative statistics by chi-square or Fisher's Exact test to determine whether the distribution of subject categories between locations a comparable or not. However, according to the methods, no such tests were applied. As far as I can tell, the author's merely did an observational comparison of their tabled data, which is insufficient.

2.) The authors stick to non-parametric sample comparison by demographic category, which is not sufficient to understand the impact of each demographic on the IgG level outcome. The authors should make an effort to apply regression analysis models to determine which demographic categories are in fact significant in the description and prediction of the outcome variable, IgG level. In this context, the authors must explain, if the level of antigen-specific IgG reflects how recent and/or frequently an individual was exposed to vector species, or if this is an artifact of the outbred model, humans, where massive data variability is to be expected due to difference in host generics.

3.) Further, the authors need to make it clear that it is MEDIAN IgG responses that are higher or lower between study sites and not IgG responses per se. E.g., Njombe has more extreme high responders than Kekem, but also more low responders. Thus, in principal the Njombe data has much higher variance than the Kekem data, which may mean something, and perhaps a slightly higher median IgG response. However, the majority of individuals are within the same range between both sites. This information is of greater interest rather than just statistically significant differences between group medians.

4.) The authors must undertake a careful analysis of their correlation plot. Eyeballing the graph suggests that the weak correlation is dependent on the weight of the 5 data points farthest to the right of the graph. A careful statistical analysis should be able to establish this. If true, this would mean that their is in fact no relationship between Anopheles and Aedes responses in the majority of individuals. Also, it would be good to know if these five data points come from the same sample site or represent the most extreme responders from all sites. Etc.

5.) There are no known negative samples on the graphs, which means that I do not know what the cutoff would be between positive samples, true responders, and negative samples, which may still have signal over background. At the moment, I must assume that everything above 0 is considered positive, meaning that every individual must have been bitten by species of both genera, assuming that the provided antigens are genera specific rather than species specific. Is that a realistic observation? It may be helpful to establish what negative samples look like. Do they provide signal above the background, too? I understand that negative samples are hard to come by considering the global distribution of both genera and the likelihood that virtually everyone has been expose to either genera at some point. I appreciate that the authors attempted to circumnavigate that problem by presenting delta ODs. Either way, it would be useful to know, if the authors, thus, interpreted anything above zero as an indicator of exposure, which brings me back to my previous comment whether the delta OD is an indicator of how recent and/or frequently someone was bitten.

Addressing these points, I believe, will significantly enrich this manuscript and improve its impact. Thus, I hope the authors will take these points under consideration when revising their manuscript as I hope to see their work published.

6. PLOS authors have the option to publish the peer review history of their article (what does this mean?). If published, this will include your full peer review and any attached files.

Reviewer #1: **Yes: **FAITH ONDITI

Reviewer #2: **Yes: **Johannes S. P. Doehl

---

## [Author Response · Author response to Decision Letter 0]

10 Nov 2024

PONE-D-24-31144

The Journal Editorial Office

 PLOS ONE

 Dear Editor-in-chief, Enclosed please find our revised manuscript, “Characterization of human exposure to Anopheles and Aedes bites using antibody-based biomarkers in rural zone of Cameroon” with the requested amendments (in red) for publication as a research article in the Journal PLOS ONE.

Editorial Requirements for Revisions

Journal Requirements:

1. Please ensure that your manuscript meets PLOS ONE's style requirements, including those for file naming. The PLOS ONE style templates can be found at https://journals.plos.org/plosone/s/file?id=wjVg/PLOSOne_formatting_sample_main_body.pdf and https://journals.plos.org/plosone/s/file?id=ba62/PLOSOne_formatting_sample_title_authors affiliations.pdf

Answer: Thanks, the manuscript has been revised accordingly.

2. In the ethics statement in the Methods, you have specified that verbal consent was obtained. Please provide additional details regarding how this consent was documented and witnessed, and state whether this was approved by the IRB.

Answer: Thanks for the question. We noted in the manuscript (line 114-121) that the verbal consent was noted on each survey in the field if the person accepted to participate in the study or not. 

Answer: Thanks for the comment. We added in the manuscript that an authorization of each local divisional officer was obtained (line 114-121). File of all signed authorization was uploaded.

4. Thank you for stating the following financial disclosure: This work received financial support from the Bill & Melinda Gates Foundation, Grant ID: OPP1210340 to C.A.-N and Pan African Mosquito Control (PAMCA) association for the field work and from “Appui à la Formation, la Recherche et à l’Innovation pour le Développement Intra-Afrique” (AFRIDI) fellowship under Intra-Africa Academic Mobility Scheme of the European Union for the laboratory analysis to I.N.N.-S. 

Answer: Dear Editor, we edited the manuscript and noted that the funders had no role in study design, data collection and analysis, decision to publish, or preparation of the manuscript. (see line 341-348).

Answer: We thank the Editor for the point addressing above. The manuscript has been revised 

6. We note that Figure 1 in your submission contain [map/satellite] images which may be copyrighted. All PLOS content is published under the Creative Commons Attribution License (CC BY 4.0), which means that the manuscript, images, and Supporting Information files will be freely available online, and any third party is permitted to access, download, copy, distribute, and use these materials in any way, even commercially, with proper attribution. For these reasons, we cannot publish previously copyrighted maps or satellite images created using proprietary data, such as Google software (Google Maps, Street View, and Earth). For more information, see our copyright guidelines: http://journals.plos.org/plosone/s/licenses-and-copyright.

 Answer: We removed Figure 1 from the manuscript and used the adjusted form of the study site figure published in our 2022 paper (Ngangue-Siewe et al., 2022).

Answer: We thank the Editor for the point addressing above. The manuscript has been revised and we are taken into account all Editorial Requirements in revised MS.

Responses to reviewer’s comments:

Please find below the responses to each reviewer comments or suggestions. Additionally, requested changes were directly introduced in red text in the revised manuscript (MS).

Reviewer reports:

Reviewer 1:

1. The study utilized ELISA to draw its conclusion. An alternative tool should have been used alongside this method in order to compare and validate these findings. In the current state, the data provided does not provide reasonable information to conclude whether this method is better than conventional methods that have been previously used (refer to introduction section lines 76–81).

Answer: Thank you for your insightful comments. We agree that using entomological data, such as human landing catches, could have helped compare exposure to Aedes and Anopheles vectors in these villages. However, the purpose of this study was not to validate these biomarkers but to use them as validated tools to evaluate spatial heterogeneity of exposure to mosquito bites in rural Cameroon. Indeed, these biomarkers of human exposure to Anopheles and Aedes mosquitoes have been developed, compared and validated in different epidemiological contexts over the past two decades (see lines 83-91, references 13-15). All these results pointed out the reliability of this immunological tools as epidemiological biomarkers of Anopheles and Ades bites. In addition, ELISA (in contrast to their immunological technic as multiplex) was the validated technology for assessing biomarkers of exposure to mosquito bites since 20 years by the IRD team, the international reference team in this topic (> 65 articles) as indicated in MS (reference 17 as review).

2. The authors did not use appropriate controls in their assay. Although the reported values are changes in OD levels, samples from positive and negative individuals (exposed and non-exposed individuals) should have been included in the assay. Similarly, the cut-off value was not determined or indicated.

Answer: We thank the reviewer for this comment. We did use a positive control to monitor plate to plate variations and make sure our experiments effectively work. This has been added in the M&M section (lines 154-158). However, we were not able to use a negative control to calculate the cut-off due to the difficulty to get serum from unexposed individuals in Africa. Indeed, the biomarkers tool is presented only in terms of specific IgG levels/intensity and not as immune responders (positive/negative). By consequences, we do not estimate the cut-off value was necessary to be determine. 

3. The study observed variation in OD values with regards to LLIN use. These antigens can last up to 40 days before they begin to wane. A follow-up study would be significant in determining this variation and would strengthen the conclusion.

Answer: We totally agree with the reviewer comment. Unfortunately, this was a cross-sectional study. While we did not have a full picture of the variation over time, we did have a picture at the time of the study (rainy season). Follow-up studies made with these biomarkers have shown that IgG responses to these peptides varied according to the season with an increase during the wet season compared to the dry one (Elanga et al., 2012 and Doucoure et al., 2013 for Aedes; Sagna et al., 2013, Poinsignon et al., 2009, and Dramé et al., 2013 for Anopheles). Follow-up could be challenging as the individual/participant could receive another bite before the 40 days.

4. Additional confounding factors were not factored in. For example, the authors do not show how they controlled for the migration of individuals across villages during the sampling period. The use of other vector control strategies like coils, indoor residual spraying, level of education/awareness of malaria control strategies, etc., at the household level was not considered, and this might influence the outcome of the study.

Answer: We appreciate your concern regarding the potential confounding factors that were not addressed in our study. We acknowledge that factors such as the migration of individuals across villages, the use of other vector control strategies (e.g., coils, indoor residual spraying), and the level of education/awareness of malaria control strategies at the household level could influence the study’s outcomes. Before starting our survey, we asked if the participant was in the study site for more than one month and concerning other vector control strategies, unfortunately, due to data availability, we were unable to control for these variables in the current study. In future research, we plan to incorporate these additional factors to provide a more comprehensive analysis, as indicated in discussion in the revised MS (lines 310-315). 

5. Data analysis

• The authors should justify why they opted for a non-parametric test instead of a parametric test.

Answer: Thanks for your question: Our data did not meet the normality assumption required for parametric tests, and the data we analyzed were ordinal in nature, which is more appropriately handled by non-parametric tests, as indicated in line 162-163. 

• The authors should consider using a violin plot or box plot in Figure 2.

Answer: We appreciate your concern regarding the type of our graphics but we preferred to use this scatter plot to better show results of specific IgG at populational level by studied group and their variation according to villages.

• What statistical test is represented in Table 2? Is it the post-hoc test or Kruskal-Wallis test? This should be reflected in line 182.

Answer: Thanks for the remark it was a Kruskal-Wallis test and we noted it in line 186 as recommended. Also, the Mann-Whitney test was used when there were only two groups to compare. 

• Supplementary File 1 should also include a comparison of the peptides with other variables such as age, gender, and the condition of bed nets.

Answer: We thank the reviewer for the suggestion but we could not compare the IgG responses to peptides with other variables such as age, gender and bed nets condition because we did not have enough data per village regarding these different variables mentioned above. For example, regarding the condition of the mosquito nets and the age of the individuals, for some villages this was not relevant because we only had 2 or 3 mosquito nets with holes, the rest being without holes and we also had few individuals in certain age groups. So, it was difficult to do a statistical analysis with this

• The correlation between the two peptides should be compared in each village, as opposed to clumping everything together (Figure 3).

Answer: A village-by-village correlation between IgG levels to the two peptides has been made and results are described in lines 241-249 in the revised manuscript. But only the Figure combining all villages is presented.

• There is no evidence of the data reported on the correlation of village-specific analysis (lines 224–29).

Answer: Dear reviewer, we have 4 villages and have used 2 peptides. A village-specific correlation will give us 8 Figures. Since many correlations by village were not significant, we decided to only provide Figure 3 representing the correction between the 2 peptides when all villages are gathered and just provide a description for the village-specific correlation (line 241 to line 249). 

• Supplementary data with all OD levels would be useful.

Answer: Thank you. The OD values have been provided as supplementary file as requested.

6. Study conclusion.

The study, though relevant, is not novel. The correlation between these findings and the infection rate would be more relevant. qPCR can be used to determine parasitemia levels in these samples.

Answer: We appreciate your insights and feedback. Our research provides a unique perspective by joining evaluation of human exposure to both Anopheles and Aedes bites in the same individual and it provides new methodology about human exposure to Aedes bites which has never been done in my country. Previous studies suggested that more mosquito bites elicit higher anti-gSG6-P1 antibodies and that higher antibody levels are also associated with a higher probability of being exposed to a bite from a Plasmodium-infected mosquito (Londono-Renteria et al., 2015 and Sagna et al., 2019). The measurement of real human-vector contact through antibody-based biomarkers could be used as a proxy to evaluate the risk of disease transmission. This is the reason why, without including information about infection status (arboviruses or Plasmodium) of the participant in this present study, we assumed that villages with children presenting high antibody responses to mosquito salivary peptides were where the risk of disease transmission were the highest.

Reviewer 2:

1. The demographic data presented in table 1 offers itself for comparative statistics by chi-square or Fisher's Exact test to determine whether the distribution of subject categories between locations a comparable or not. However, according to the methods, no such tests were applied. As far as I can tell, the author's merely did an observational comparison of their tabled data, which is insufficient.

Answer: We thank the reviewer for this for your insightful feedback. We acknowledge that the initial submission did not include the application of comparative statistical tests, such as the chi-square, to determine the comparability of subject categories between locations. In response to your valuable suggestion, we have now applied the chi-square test to the demographic data in Table 1. The Table 1 reflects the p-values and Chi-squared values obtained from this test, providing a more rigorous comparison of the demographic data.

2. The authors stick to non-parametric sample comparison by demographic category, which is not sufficient to understand the impact of each demographic on the IgG level outcome. The authors should make an effort to apply regression analysis models to determine which demographic categories are in fact significant in the description and prediction of the outcome variable, IgG level. In this context, the authors must explain, if the level of antigen-specific IgG reflects how recent and/or frequently an individual was exposed to vector species, or if this is an artifact of the outbred model, humans, where massive data variability is to be expected due to difference in host generics.

Answer: We cannot do logistic regression for the simple reason that we don’t have binary data (positive vs negative) concerning the biomarker results. If we had a negative control and calculated the positivity threshold, we could have done it.

3. Further, the authors need to make it clear that it is MEDIAN IgG responses that are higher or lower between study sites and not IgG responses per se. E.g., Njombe has more extreme high responders than Kekem, but also more low responders. Thus, in principal the Njombe data has much higher variance than the Kekem data, which may mean something, and perhaps a slightly higher median IgG response. However, the majority of individuals are within the same range between both sites. This information is of greater interest rather than just statistically significant differences between group medians.

Answer: Thank for your comments. We have revised the manuscript to emphasize that it is well the median IgG responses that differ between Njombe and Kekem, rather than th

---

## [Editor Report · Decision Letter 1]

15 Nov 2024

Characterization of human exposure to Anopheles and Aedes bites using antibody-based biomarkers in rural zone of Cameroon

PONE-D-24-31144R1

Dear Dr. Ngangue-Siewe,

We’re pleased to inform you that your manuscript has been judged scientifically suitable for publication and will be formally accepted for publication once it meets all outstanding technical requirements.

Kind regards,

Nazarudin Safian, PhD

Academic Editor

PLOS ONE
---

## [Editor Report · Acceptance letter]

26 Nov 2024

PONE-D-24-31144R1 

PLOS ONE

Dear Dr. Ngangue-Siewe, 

I'm pleased to inform you that your manuscript has been deemed suitable for publication in PLOS ONE. Congratulations! Your manuscript is now being handed over to our production team.

Kind regards, 

on behalf of

Professor Nazarudin Safian 

Academic Editor

PLOS ONE